# Matrix of Affordable Housing Assessment: A Development Process

**Afaq Hyder Chohan**

Department of Architecture, College of Architecture, Art & Design, Ajman University,
Ajman P.O. Box 346, United Arab Emirates; a.chohan@ajman.ac.ae; Tel.: +971-55-3880771

**Abstract:** The United Arab Emirates (UAE) is a multiracial society with diverse housing and a potential real estate market. This study focused on users' perceptions of the designs of available and affordable private housing stock in Dubai, Sharjah, and Ajman, which are the most populated states (emirates) of the UAE. A literature review and case studies of low- to medium-rise residential buildings were used to determine the parameters defining affordable housing design, and a model was developed of 7 design segments (independent variables) with 39 dependent variables. The model consists of a matrix of 39 design variables, in which each variable is set in a survey tool with a Likert scale to evaluate user satisfaction levels with the designs of their respective buildings. Questionnaires were distributed among the inhabitants of several buildings at different locations in the emirates. This study found that 16 anomalous design factors failed to satisfy users. It is likely that the results of this study will provide a blueprint for dialogue between regional building designers and end users to improve the designs of new buildings. The resulting design assessment matrix can be used for the analysis of residential buildings in other parts of the Gulf Cooperation Council region.

**Keywords:** affordable housing assessment; building defects; real estate assessment

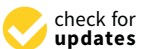



## 1. Introduction

The United Arab Emirates (UAE) is an important member of the Gulf Cooperation Council (GCC). It is a unique country because of its multinational, harmonious culture. It is home to people from 180 countries, and this fact is a driving force for the growth of its economic and real estate markets [1]. Many countries have recorded slow or negative economic growth because of the COVID-19 pandemic; however, the UAE has seen steady growth. In general, people from neighbouring Asian countries consider it to be a safe and secure option for investment [2]. Stable economic growth and development always increase the demand for comfortable living and working environments [3]. The UAE's multicultural society and population influxes set high demands for diverse types of housing and design solutions. Dubai is a major city in the UAE and defines standards and trends for economic growth. Statistics show that during 2020, the real estate market had a supply of 690,498 housing units (apartments and villas) in Dubai alone, whereas the overall growth in housing demand in the UAE is approximately 50,000–60,000 housing units per year [4]. Increasing costs of materials, and construction and global slowdowns affect housing affordability. Affordability is a term for affordable housing or social housing introduced to define the type of housing intended for less well-off sections of the population [5,6]. In the United Kingdom and some other European countries, affordable housing refers to housing for specific income groups, defined by the national government. The threshold of affordability varies from region to region, but affordable housing is generally classified as residences for low- to middle-income groups of society [7]. In developing countries, the concept of affordable housing is defined as basic houses for economically weaker sectors, or homes at affordable prices for low-income groups [8]. The USA Federal Department of Housing and Urban Development (HUD) has provided a comprehensive definition of affordable



housing. It defines an "affordable dwelling" as a house rented, or mortgaged, for 25–30% of the resident's monthly income [9].

The rule of thumb is that housing is affordable if low-income households spend less than 30% of their income on housing. Therefore, mass housing projects are affordable, provided they meet the affordability criteria/requirements [10]. The concept of the "Grow Home" is equated with affordable housing. In principle, the concept was established to encourage families to have their own homes [11]. Liveable, pleasant housing is a fundamental requirement for people. Two factors are important for making housing affordable for most people. First, it should not be luxurious; second, it should not be used for a commercial activity [12].

A higher housing demand continuously challenges designers to create an optimum design. New and untested design initiatives usually result in several different maintenance issues, and designers do not always identify these factors at the design stage. However, users of such designs usually realise the shortcomings of the design process [13,14]. Building houses is a complex design process that results from layer-by-layer information on building components. Each building component must be designed with the future in mind and an understanding of its relationship with other building components [15]. Housing design includes urban planning, social aspects of technology, environmental considerations, and other issues related to development. A basic house should allow couples to structure their families and ensure accommodation. A basic residence should have habitable spaces to allow users to carry out established functions, such as living rooms, dining rooms, kitchens, bathrooms, bedrooms or utility rooms, space conditioning, window arrangements, and accessibility [16].

Deficient, untested designs of building components can cause problems and deterioration and thus reduce the functionality of forethought design [17]. Building design elements that cause defects and require frequent maintenance not only affect the residential unit; the entire project can be disregarded in the real estate market. This phenomenon results in dissatisfaction among occupants [18].

As in other engineering fields, the design stage of built environment projects is critical for the success of a product. Imprecise communication among stakeholders, such as designers, clients, and builders, is a major reason for deficient design [19]. Information on intended use and users' behaviour is important in the design stage; the designer often relies on simulations, which are not the true perceptions of real users. This disagreement at the design stage often causes challenges in the occupational stage [20–22]. However, in the field of built environments, project completion takes a much longer time than project design. Miscommunication during project construction, mismanagement, and poor workmanship often cause gaps in the design implementation and result in deficient design of building components [23,24]. However, incomplete drawings and material specifications are important factors that can cause building defects [21,25]. Lack of engineering experience, poor information on new materials, and deficient construction skills are also contributing factors in building component faults [26,27].

Defects are not limited to buildings in developing countries; their occurrences are profound in developed countries such as New Zealand, the United Kingdom (UK), Spain, and Australia [28]. The UK construction industry is inundated by building defects [29,30]. In addition, Mills et al. [31] and Love and Sohal [32] revealed that residential buildings are more vulnerable to defects, and their repair and maintenance comprise an enormous part of household budgets. Buildings weaken and depreciate rapidly with the frequency of defects, and annual maintenance costs can reach up to 4% of the total construction cost [33]. Users' health issues are also often related to different building service defects. Only through design appraisal of a building's existing condition and the level of damage can they be determined [34,35]. It is important to conduct such a building assessment before the repair and maintenance stages. Early diagnosis of building defects and their repair will eventually enhance the building life cycle and help to maintain the real estate value [34]. Building appraisal or condition surveys are activities that start with participation and physical

scrutiny. A list of main defects is developed, if required, with details in the sub-section. Defects are verified and possibly used as variables to evaluate the existing conditions of a building [36].

To summarise, the UAE is a habitat of diverse communities and nationalities, and each community has its choice of housing style. Therefore, the real estate market of the UAE has a variety of housing design solutions. Early studies insist that possible causes of building defects include miscommunication, value engineering, project management, and new and untested design concepts, besides issues of construction. Moreover, the use of untested concepts for designing building components initiates maintenance issues in the post-occupational stage. Defects and their implications are onerous for occupants, depreciate the value of their residential unit, and eventually affect the real estate market rating of the entire project. Early detection of building defects can stop these defects and maintenance phenomena. This is only possible through an impartial and truthful analysis of building assessment through various stakeholders of building design processes, that is, designers, construction engineers, contractors, and end users.

The discussion of building defects considers aspects of design quality and its practice during building design [14,28,37]. This study also regards building defects as parameters of building design, and their presence and frequency in different building components help to rate building design quality. Defects can appear both inside and outside a building. For example, defects inside can significantly affect the environmental conditions of spaces. Defects outside affect aesthetics and physical conditions both outside and inside the building.

The aim of this study was to develop a defect- and design-based building assessment model to evaluate users' perceptions of (performance) design factors and defects in their respective houses. This model encompasses design factors, physical defects, and environmental issues of the building. Therefore, the commissioning detail model of design assessment consists of seven segments of building design, as shown in Figure 1. This model is used to capture a detailed assessment report to determine maintenance issues and highlight design factors that require improvement in future designs. In addition, it helps to proactively take suitable repair measures based on the condition assessment of buildings. Adaptation of this matrix will ensure accurate evaluation of the condition of a building (inside and outside) and can reduce time and cost.

Previous studies [29,38–40] have also developed and used building assessment models that evaluate building defects. These models provide an inclusive condition assessment of buildings that can help stakeholders in decision making, maintenance budget allocation, repair, and rehabilitation of existing buildings.

Other studies have highlighted the fact that most roof defects result from deficient roof design, due to imperfect roofs, where moisture and water penetration affect the structural elements and internal environment of the building [41]. Most buildings suffer from different types of construction defects; however, roofing defects have the highest percentage (22%) of all defects [42,43]. Suitable material selection, proper stability testing, and ample knowledge of the properties of the materials are important factors for determining the durability and life cycle of a building [44]. This is particularly important when introducing new materials for building façades because building façades are highly vulnerable to defects [45,46].

Defects in the vertical and horizontal structural elements are crucial for determining building design and strength assessments because defects in these elements are likely to show a slow process of building failure [47]. Loadbearing masonry works as both vertical and horizontal structural elements in buildings and always suffers from defects related to bonding materials and workmanship. Defects appear as aggressive dampness and the resultant degradation of the wall system [48]. In addition, low-quality masonry is also subject to loss of tensile and compressive strength, porosity, and frostbiting [49,50].

In addition to the demand for aesthetically appealing façades, building envelopes are responsible for overall thermal comfort [43]. Well-designed wall systems ensure durability, safety, and comfort in buildings and are significant influencing factors of user

satisfaction [51]. The building envelope plays a fundamental role in creating a barrier between the atmospheric environment and building users [52–54]. However, environmental loads such as extreme temperatures, frost, rainfall, humidity, and wind strongly affect the façade surfaces. Moreover, these loads initiate deterioration and shorten the life cycle of the façade, which result in expensive maintenance and a decline in the market value of buildings [55–58].

## 2. Materials and Methods

### 2.1. Study Design

Through participation at several different sites (case studies), different levels of defects and their context in building components were reviewed. Eventually, a residential building assessment model (RBAM) was developed to evaluate affordable housing/building design through user perception in the UAE, as shown in Figure 1.

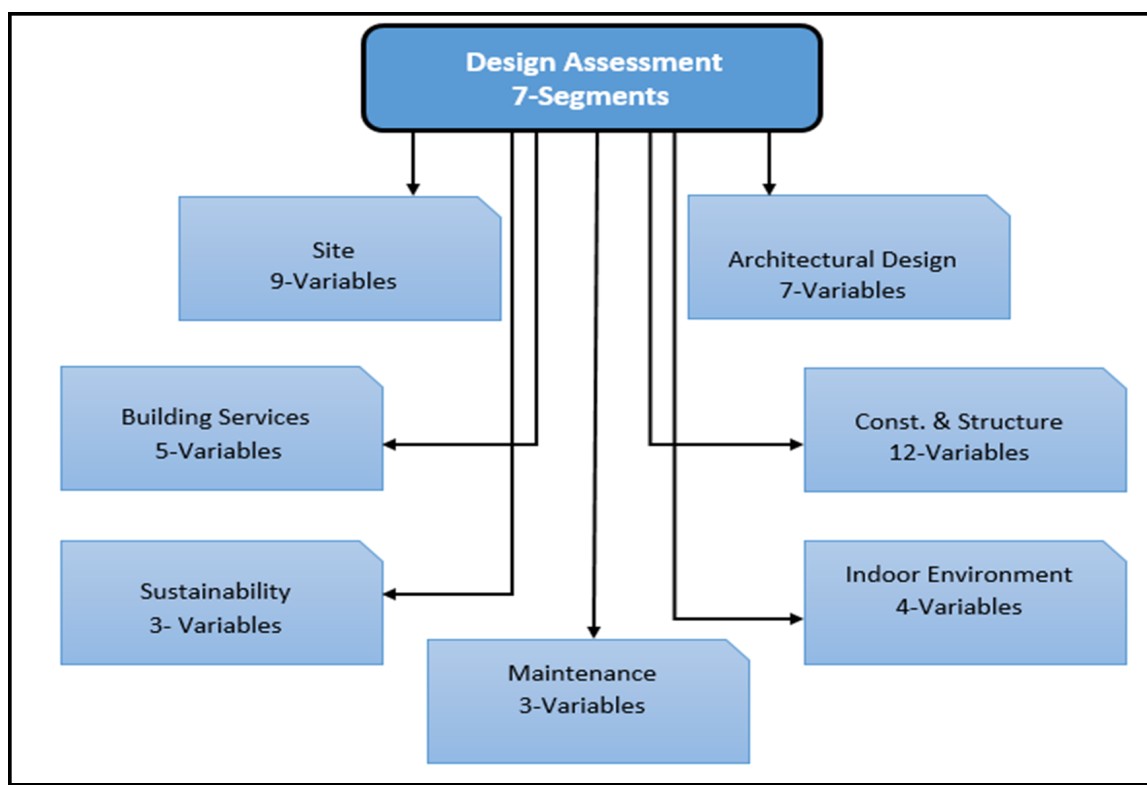

**Figure 1.** UAE residential building assessment model (RBAM).

The model developed is flexible and permits continuous revision to include/reject different types of design deficiencies or resultant building defects for future studies. Each segment of the model was assigned different dependent variables to determine users' perceptions of building design, as shown in the matrix in Table 1. In addition, this matrix could be useful for design evaluation in other regions of GCC countries because of similarities in design and construction techniques, climate, culture, and social issues. A comparative analysis of the UAE-RBAM model in Figure 1 revealed that earlier studies have also utilised similar building design factors to assess building conditions. Table 2 shows the outcomes of similar studies in other parts of the world.

The literature review and participation at different residential locations in the UAE helped break down each segment of the RBAM model into a matrix of 39 design variables representing the different building components, as shown in Table 1. The matrix also depicts the concentration of building design issues in Dubai, Sharjah, and Ajman.

**Table 1.** Building design assessment matrix.

| No | Design Segments and Variables | Dubai | Sharjah | Ajman |
|----|---|---|---|---|
| **A** | **Housing Site** | | | |
| 1. | Landscaping and view of housing | √ | √ | × |
| 2. | Well-designed spaces/streets | √ | × | √ |
| 3. | Sufficient parking provision | × | × | × |
| 4. | Social spaces for interaction | √ | √ | × |
| 5. | Access to popular modes of transport | √ | √ | × |
| 6. | Access to nearby public parks and schools | × | × | × |
| 7. | Availability of nearby medical facilities | √ | √ | √ |
| 8. | Pleasant neighbourhood around the site | √ | × | × |
| 9. | Streetlights and surveillance | √ | √ | √ |
| **B** | **Architectural Design** | | | |
| 1. | Distinctive identity through architectural style | √ | × | × |
| 2. | Pleasant and innovative building façades | √ | × | × |
| 3. | Standard sizes of bedrooms, kitchen, and hall | √ | √ | √ |
| 4. | Bathroom space in context of its functions | √ | √ | √ |
| 5. | Size of staircases for easy movement and emergency purposes | √ | √ | × |
| 6. | Protective building elements such as projections and overhangs | × | × | × |
| 7. | Flexibility for extension/alteration within available space | × | × | × |
| **C** | **Structure and Construction** | | | |
| 1. | Building use under natural calamity (rain, storms, and earthquakes) | √ | √ | √ |
| 2. | Construction quality for performance and aesthetics | √ | √ | × |
| 3. | Structure quality in context of hairline cracking, thermal expansion, etc. | × | √ | × |
| 4. | Quality of finishing material for bathroom internal surfaces | √ | √ | √ |
| 5. | Quality of material used for doors and windows | √ | × | × |
| 6. | Quality of masonry/plaster (dampness, cracks, etc.) | × | × | × |
| 7. | Quality of sanitary fixtures and appliances | √ | √ | √ |
| 8. | Quality of electrical work and fixtures | √ | √ | √ |
| **D** | **Building Services** | | | |
| 1. | Well-designed refuse collection system | √ | √ | √ |
| 2. | Easy to manage and maintain electrical installations | √ | √ | √ |
| 3. | Sufficient number of electrical points given in space | × | × | × |
| 4. | Sufficient water supply and storage | √ | √ | √ |
| 5. | Sufficient provision of vertical transport (elevators) | √ | × | × |
| 6. | Facility air conditioning system | √ | √ | √ |
| **E** | **Indoor Environment and Comfort** | | | |
| 1 | Summer over heating | √ | √ | √ |
| 2 | Air quality in context of humidity and moisture content | × | × | × |
| 3 | Noise from outside | √ | × | × |
| 4 | Natural ventilation in bathrooms and kitchen | × | × | × |

**Table 1.** *Cont.*

| No | Design Segments and Variables | Dubai | Sharjah | Ajman |
|---|---|---|---|---|
| A | **Housing Site** | | | |
| F | **Housing Maintenance** | | | |
| 1 | Availability of building maintenance measures/provisions | × | × | × |
| 2 | Effectiveness of maintenance | × | × | × |
| 3 | Response to maintenance request and quality | √ | × | × |
| G | **Sustainability** | | | |
| 1 | Electricity saving design techniques | × | × | × |
| 2 | Use of low-flow water fixtures and flushing cisterns | × | × | × |
| 3 | Use of recycled/recyclable materials in buildings such as glass, plastics, plaster board, wood, and industrious wood | × | × | × |

**Table 2.** Literature review: design assessment model.

| No | Research and Year | Outcome |
|---|---|---|
| [52] | Cláudia Ferreira (2021) | Study revealed that building façades are vulnerable to extreme environmental loads, and these loads initiate deterioration and minimise the life cycle of façades. Therefore, this results in expensive maintenance and a decline in market value. |
| [29] | Faisal Faqih and Tarek Zayed (2021) | Study found that many building defects are related to structure, construction, and building services of buildings. |
| [33] | F. Faqih. et al. (2020) | Fault of any design element, component, or part of a building has potential to affect users' safety, comfort, and health. |
| [34] | R. Kuijper et al. (2019) | Research developed an inclusive building condition assessment model, assisting building stakeholders in decision making, maintenance budget allocation, repair, and rehabilitation of facilities. |
| [53] | M. Buberwa et al. (2017) | Study established that housing design quality can be determined through site and layout, landscaping, unit size, users' comfort, sustainability, accessibility, and visual impact. |
| [3] | Peng Mao et al. (2017) | Study defined the defects linked to the internal environment and their effects on health of users. |
| [19] | A. Aissani et al. (2016) | Study worked out various factors relevant to the internal environment and considered that thermal comfort (heating and cooling) of building spaces is important and concerned with health of users. |
| [11] | A.H. Chohan et al. (2015) | Developed housing design quality indicators and considered factors of site selection, construction, architectural design, and building services as key indicators of design quality. |
| [39] | N.L. Othman et al. (2015) | Poor workmanship and improper waterproofing are factors that contribute to the moisture and dampness problems in building internal environments, in addition to affecting overall building functionality. |
| [18] | A.C. Menezes et al. (2012) | Factors of safety and sustainability of existing buildings are important for building condition assessment, besides being vital for users' wellbeing. |
| [10] | N.H. Ishak et al. (2007) | Presented a set of inclusive literature reviews and defined various aspects of deficient design implicating building maintenance. |
| [54] | Josep Maria Montaner, Zaida Muxi, and David H Falagan (2011) | Study presented a holistic approach to understand design of basic housing and considered spaces such as living rooms, dining rooms, kitchens, bathrooms, bedrooms or utility rooms, space conditioning, window arrangements, and accessibility as essential to allow users to carry out established functions of daily life. |
| [55] | Ivana Brkanić (2017) | Study explained housing quality in four domains, i.e., apartment unit quality criteria, apartment building quality criteria, neighbourhood quality criteria, and social and economic criteria. |

The matrix in Table 1 provides instant information about the assessment of existing designs of low- to medium-rise reinforced cement concrete (RCC) buildings at different affordable locations of the three emirates. The initial data were collected through a literature review, site visits, participation, and unstructured interviews. The content of the matrix is set in the survey tool to capture the in-depth role of the variable and the response of users towards these variables. A five-point Likert scale with 1–5 ranking was scored as not satisfied (1), not sure (2), slightly satisfied (3), fairly satisfied (4), and extremely satisfied (5), as shown in Table 3.

**Table 3.** Housing design assessment tool.

| Housing Design Assessment Tool | | | | | | |
|---|---|---|---|---|---|---|
| **Q No** | **Design Assessment Criteria** | **Users' Response to Variables** | | | | |
| A | Are you satisfied with the following factors of site related to your housing? | Strongly Satisfied | Fairly Satisfied | Slightly Satisfied | Not Sure | Does Not Satisfied |
| 1. | Landscaping and view of housing | | | | | |
| 2. | Well-designed spaces/streets around housing blocks | | | | | |
| 3. | Sufficient parking provision | | | | | |
| 4. | Social spaces for interaction | | | | | |
| 5. | Access to popular modes of transport | | | | | |
| 6. | Access to nearby public parks and schools | | | | | |
| 7. | Availability of nearby medical facilities | | | | | |
| 8. | Pleasant neighbourhood around the site | | | | | |
| 9. | Streetlights and surveillance | | | | | |
| B | Are you satisfied with the following factors of architectural design of housing? | Strongly Satisfied | Fairly Satisfied | Slightly Satisfied | Not Sure | Does Not Satisfied |
| 1. | Distinctive character in neighbourhood through architectural style | | | | | |
| 2. | Pleasant and innovative building façades | | | | | |
| 3. | Standard sizes of bedrooms, kitchen, and hall | | | | | |
| 4. | Bathroom space in context of its functions | | | | | |
| 5. | Size of staircases for easy movement and emergency purposes | | | | | |
| 6. | Conventional protective building elements such as projections, overhangs, and cornices | | | | | |
| 7. | Flexibility for extension/alteration within available space | | | | | |
| C | Are you satisfied with the following factors of structure and construction at housing? | Strongly Satisfied | Fairly Satisfied | Slightly Satisfied | Not Sure | Does Not Satisfied |
| 1. | Building use under natural calamity (rain, storms, and earthquakes) | | | | | |
| 2. | Construction quality for performance and aesthetics | | | | | |
| 3. | Structure quality in context of hairline cracking, thermal expansion, joints, etc. | | | | | |
| 4. | Quality of finishing material for bathroom internal surfaces | | | | | |
| 5. | Quality of material used for doors and windows | | | | | |

**Table 3.** *Cont.*

| | Housing Design Assessment Tool | | | | | |
|---|---|---|---|---|---|---|
| **Q No** | **Design Assessment Criteria** | **Users' Response to Variables** | | | | |
| 6. | Quality of plaster rendering both internal/external | | | | | |
| 7. | Quality of sanitary fixtures and appliances | | | | | |
| 8. | Quality of electrical work and fixtures | | | | | |
| D | Are you satisfied with the following factors of building services in housing? | Strongly Satisfied | Fairly Satisfied | Slightly Satisfied | Not Sure | Does Not Satisfied |
| 1. | Well-designed refuse collection system | | | | | |
| 2. | Easy to manage and maintain electrical installations | | | | | |
| 3. | Sufficient number of electrical points given in space | | | | | |
| 4. | Sufficient water supply and water storage | | | | | |
| 5. | Sufficient provision of vertical transport (elevators) | | | | | |
| 6. | Facility air conditioning system | | | | | |
| E | Are you satisfied with the following factors of indoor environment and comfort in the housing? | Strongly Satisfied | Fairly Satisfied | Slightly Satisfied | Not Sure | Does Not Satisfied |
| 1 | Summer over heating | | | | | |
| 2 | Air quality in context of humidity and moisture content | | | | | |
| 3 | Noise from outside | | | | | |
| 4 | Natural ventilation in bathrooms and kitchen | | | | | |
| F | Are you satisfied with the following factors of maintenance in housing? | Strongly Satisfied | Fairly Satisfied | Slightly Satisfied | Not Sure | Does Not Satisfied |
| 1 | Availability of building maintenance measures/provisions | | | | | |
| 2 | Effectiveness of maintenance | | | | | |
| 3 | Response to maintenance request and quality | | | | | |
| G | Are you satisfied with the following factors of sustainability in housing? | Strongly Satisfied | Fairly Satisfied | Slightly Satisfied | Not Sure | Does Not Satisfied |
| 1 | Electricity saving design techniques | | | | | |
| 2 | Use of low-flow water fixtures and flushing cisterns | | | | | |
| 3 | Use of affordable and available local maintenance | | | | | |

Table 1 was extracted from the literature review highlighted in Table 2 and further filtered and consolidated through participation on site. There were 36 dependent variables nested in 7 segments of independent variables. Some of these highly visible issues are discussed below.

Regarding issues of site (section A), Figure 2 confirms that parking is a critical issue, and its availability to users is not always ensured. Landscaping, well-designed streets, and pleasant neighbourhoods are sporadic in many localities. Lack of parking further narrows down the streets, and users encroach on private land for the purpose of parking.

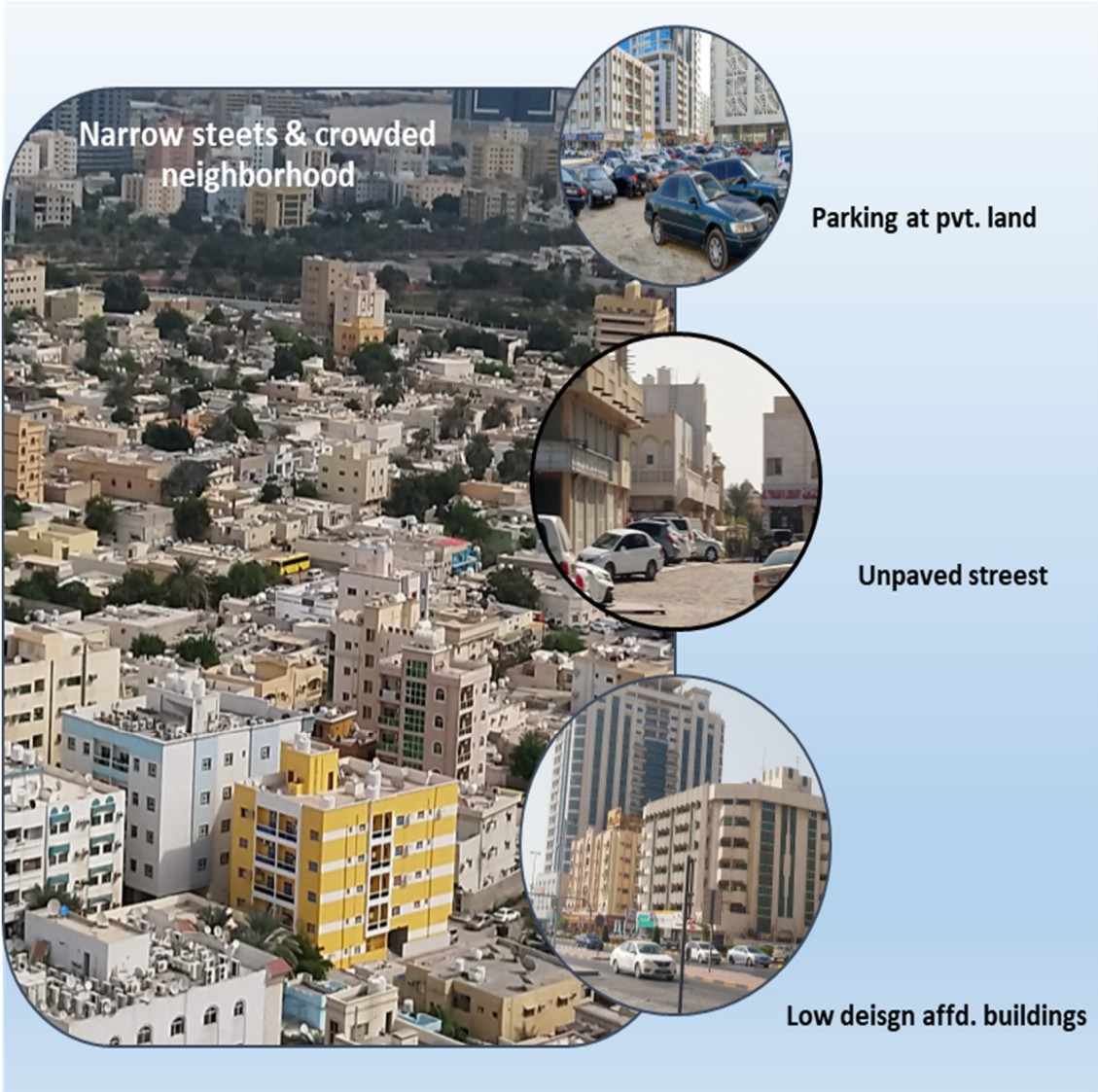

**Figure 2.** Site and neighbourhood issues.

However, some issues of architecture (section B) can be reviewed through Figures 2 and 3. However, other issues such as space size and flexibility of spaces were determined by users' perceptions. Figure 2 shows that there is a significant difference in the architectural style of different affordable buildings, which may create dissatisfaction among building users.

Regarding issues of construction and structure (section C), Figure 3 depicts serious issues of cracking, decaying finishing materials, and reinforcement failure. Many buildings have experienced such issues after a few years of occupation. These issues are serious to users and lead to buildings deteriorating earlier than expected, causing immense anxiety about safety and resale value.

Figure 4 shows some of the issues mentioned in section D of the matrix and their implications for users. Periodic maintenance is important to users at the post-occupational stage, and frequent maintenance and failure of building components causes stress and frustration among users, particularly residents of medium-rise and high-rise buildings who may become highly annoyed by the performance of elevators.

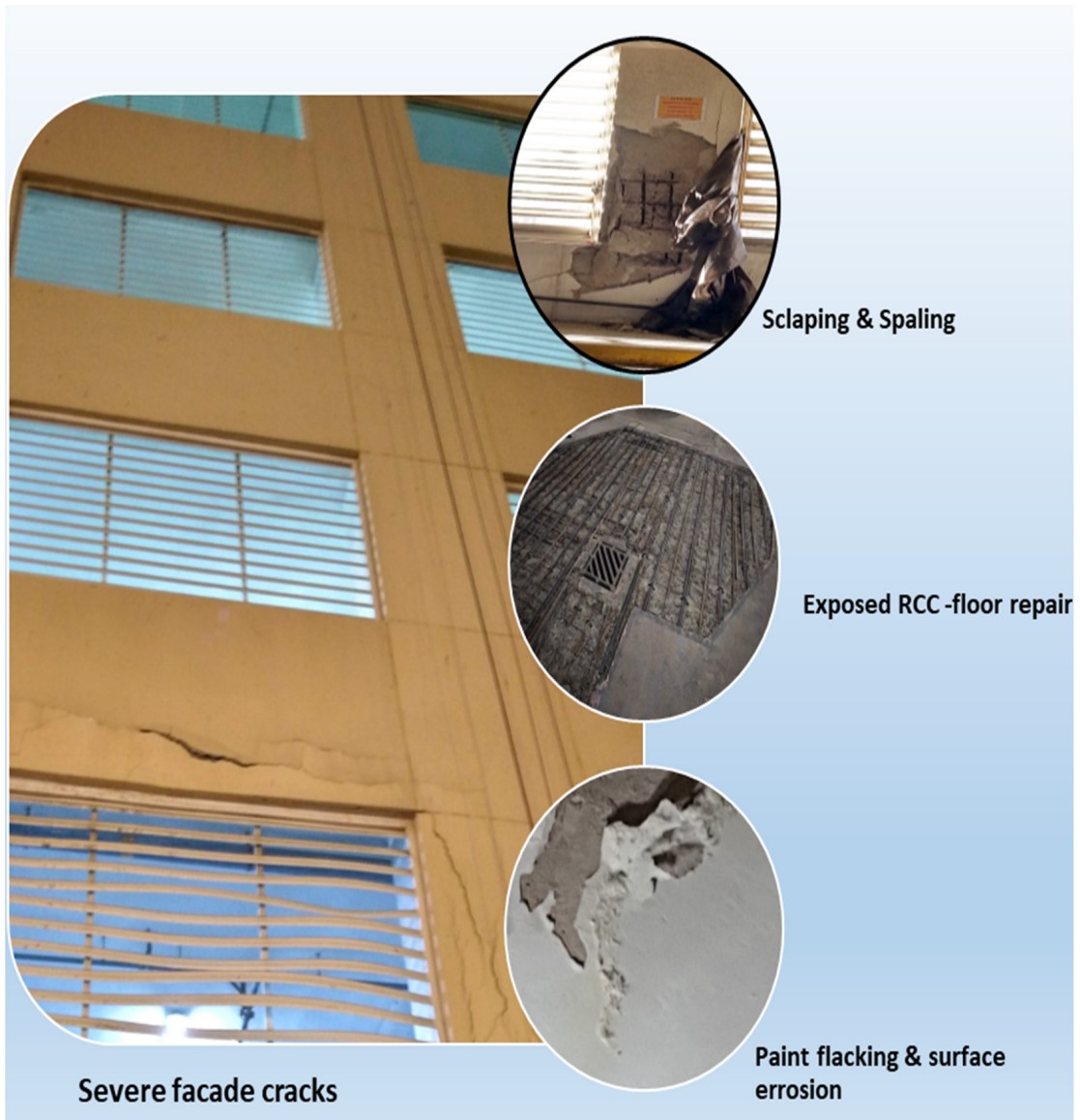

**Figure 3.** Construction and structural defects.

The indoor environment section (E) of the matrix is related to variables that require the direct response of users and cannot be directly illustrated. However, the defects visible in Figure 4 also affect the indoor environment and air quality. These issues are relevant to affordable housing and are compromised at different locations.

In addition, the variables mentioned in sections G and H are highly relevant in affordable housing. Because of the low-budget design, different aspects of building maintenance and sustainability are often neglected. Many users of buildings in the study were unhappy with the quality and response time of maintenance work. Maintenance is a highly understandable phenomenon and constitutes the usual immediate response to users' queries. Photographs in Figure 4 also show that a significant amount of maintenance work results from a poor or deficient design.

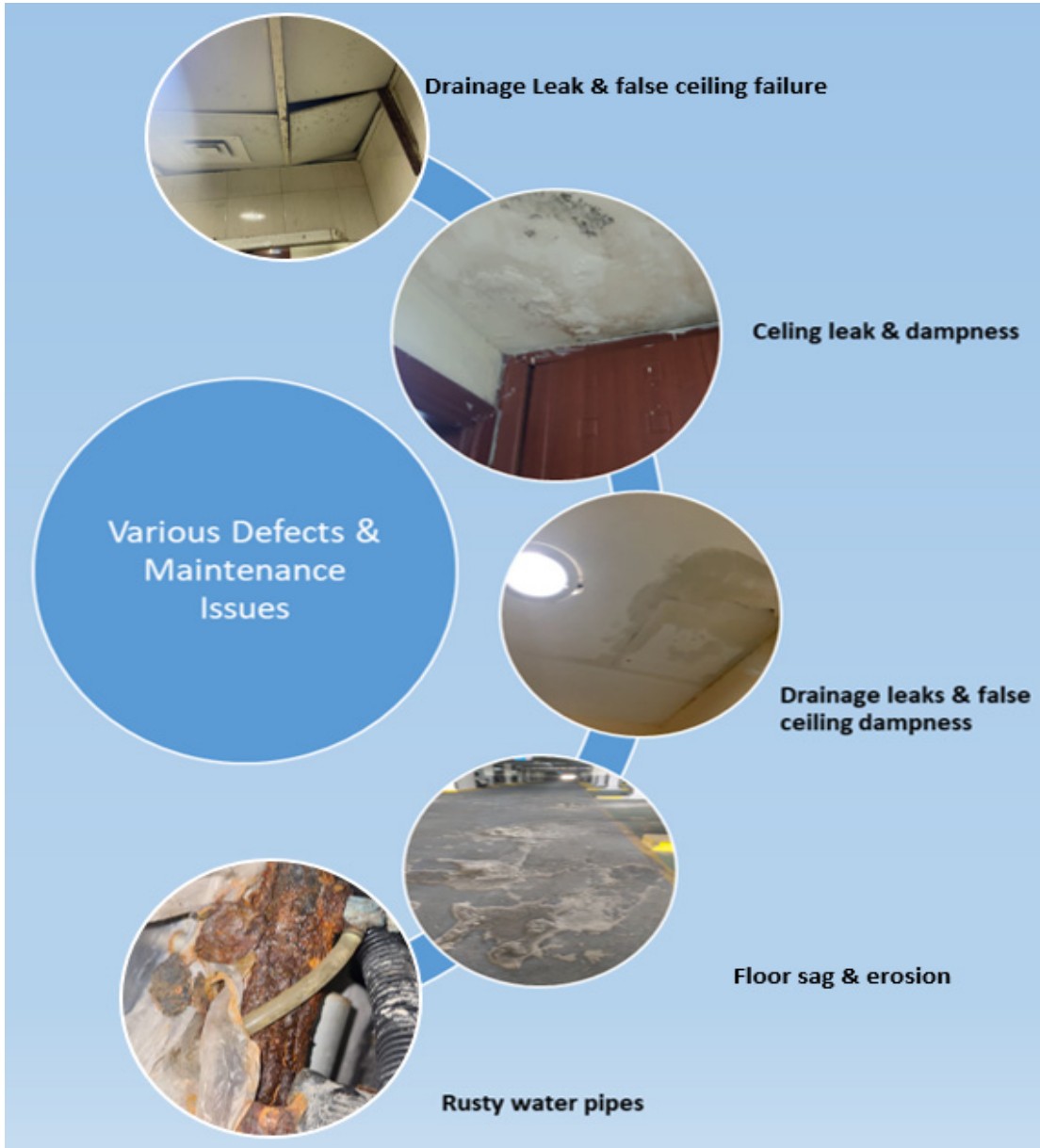

**Figure 4.** Different defects and maintenance issues.

Sustainability is very broad and emerging, and not all aspects of sustainability are relevant to the project under study, particularly in buildings aged 10 years and older. Therefore, only basic issues of sustainability were investigated in the matrix, specifically those issues that directly affect the users' monthly budget such as water and electricity consumption.

The study was planned to add new information about the correlation between various design variables and users' satisfaction with affordable housing in the UAE. Moreover, the existing gap in appropriate (quality) building design and defects was investigated. The key research objectives were as follows:

1.  To identify the percentage of satisfied or unsatisfied building users in Dubai, Sharjah, and Ajman;
2.  To identify which building design variables users were satisfied and dissatisfied with;
3.  To determine the gap between acceptable (quality) design and dissatisfaction factors.

*2.2. Methodoloy*

The building design assessment methodology was planned based on the classic technique of site participation, informal interviews, and a quantitative questionnaire survey. A three-layered investigation was set to complete this research: The first layer established issues of deficient and unsatisfactory building design through a review of the literature and participation on site (Table 1).

The second layer produced a tangible questionnaire survey (based on identified issues in Table 3) which was distributed among different locations of affordable housing in Dubai, Sharjah, and Ajman. In the third layer, the quantitative data obtained were analysed using SPSS computational software, and the results are shown in Figures 5–8.

A questionnaire with closed-ended questions was designed to fulfil objective 2 of the study and to capture the level of satisfaction or dissatisfaction with the liability of 39 building design variables. This tangible data collection method is reliable and time friendly, and the data obtained through this method are consistent, clear, and reliable [14,18,22].

The design survey tool (questionnaire) of the study was grouped into seven segments (A–G), with segments A, B, C, and D concerning design criteria of buildings and sections E, F, and G that are relevant to the building environments and facility/estate management and maintenance. Care was taken to keep the questions easy to understand; however, some questions were explained further with keywords within the question: for example, C3—Structure quality in the context of hairline and thermal expansion cracks.

Building users from different social and monetary backgrounds served as survey respondents and were required to answer all the questions regardless of their status and sections of the questionnaire. Arabic is the first language of the region; therefore, for the convenience of some respondents, the questionnaire was translated and distributed in Arabic. Sample sizes > 30 and < 500 are appropriate for most research; however, for a single type of respondent, a minimum size of 30 samples is necessary [59,60].

## 3. Results and Discussion

A total of 210 questionnaires were distributed among residents at different locations of affordable housing split evenly between the three emirates (70 questionnaires in each emirate). The quantitative survey tool worked well, and 92% of respondents replied to all questions; however, 8% of respondents in some questions chose the "not sure" option. A total of 123 respondents, 58% (approximate) of users from the three emirates, delivered their feedback in a timely manner.

The data obtained from the survey were analysed as "mean" (significance) and "analysis". Table 4 lists the 16 significant issues identified by the respondents. This analysis is based on the consent of respondents about different issues judged through the notion of a higher mean value. During analysis, this study found that for some issues, a high number of respondents chose scales "not satisfied = 1" and "slightly satisfied = 3", but their responses were divided and did not reach a significant level. Therefore, to ascertain the authentic poor design issues, this study considered the summation of both extreme scales of dissatisfaction.

**Table 4.** Issues of poor design in affordable housing.

| **Housing Site** |
|:---:|
| Landscaping and view of housing |
| Well-designed spaces/streets |
| Sufficient parking provision |
| Social spaces for interaction |
| Access to popular modes of transport |
| **Structure and Construction** |
| Construction quality for performance and aesthetics |
| Structure quality in context of cracks |
| Quality of internal plaster rendering/ surfaces |
| **Building Services** |
| Quality of electrical works & fixtures |
| Facility air conditioning system |
| **Indoor Environment and Comfort** |
| Summer over heating |
| Air quality in context of humidity and moisture content |
| Natural ventilation in bathrooms and kitchen |
| Building Maintenance |
| Availability of building maintenance measures/provisions |
| Maintenance of drainage system |
| Building Sustainability |
| Use of low-flow water fixtures and flushing cisterns |

Figure 5 nests the variables that reached the threshold of dissatisfaction. Besides this, some variables at certain locations showed significant satisfaction. This response is due to the survey being conducted at different locations and in different emirates of the UAE. It is significant to note that different affordable buildings were assessed through the matrix (Table 1), and not a single architectural design variable in the matrix reached the threshold of significant dissatisfaction. This indicates that architectural design in the UAE, particularly in the emirates of Dubai, Sharjah, and Ajman, is being practiced assiduously, and users showed contentment with the design function and performance at the post-occupation stage.

However, users showed moderate to high discontentment about some issues related to segments of site (five issues), construction and structure (three issues), building services (two issues), internal environment (three issues), maintenance (three issues), and building sustainability (one issue).

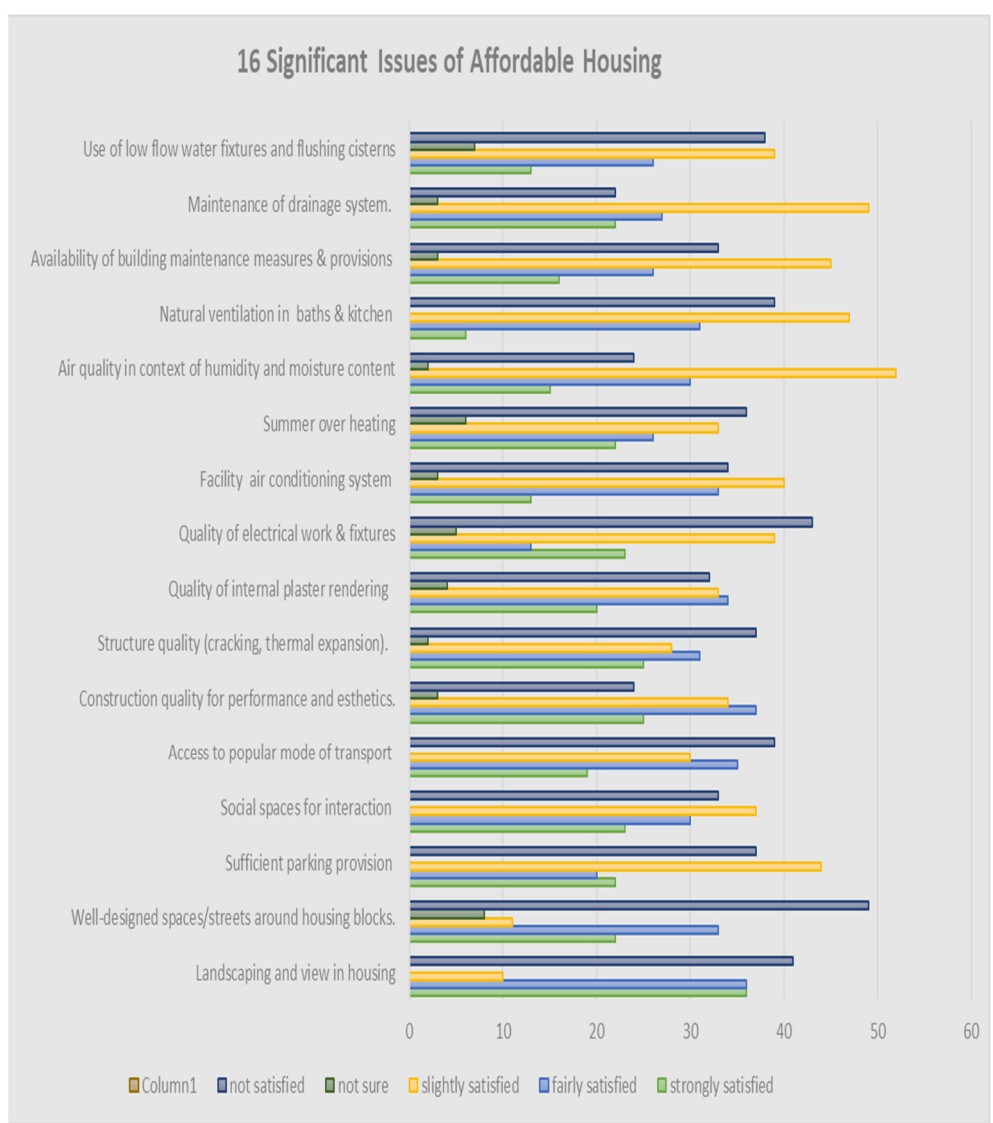

**Figure 5.** Significant affordable housing issues.

Figure 6 above shows the trend for "not satisfied" responses for different variables listed in the matrix. It is worth mentioning that the highest number of users were dissatisfied with nine variables such as landscaping and view of housing, well-designed spaces/streets around housing blocks, access to popular modes of transport, structure quality in the context of hairline cracking and thermal expansion, quality of electrical work and fixtures, summer overheating, natural ventilation in bathrooms and kitchen, availability of building maintenance measures and provisions, and use of low-flow water fixtures and flushing cisterns.

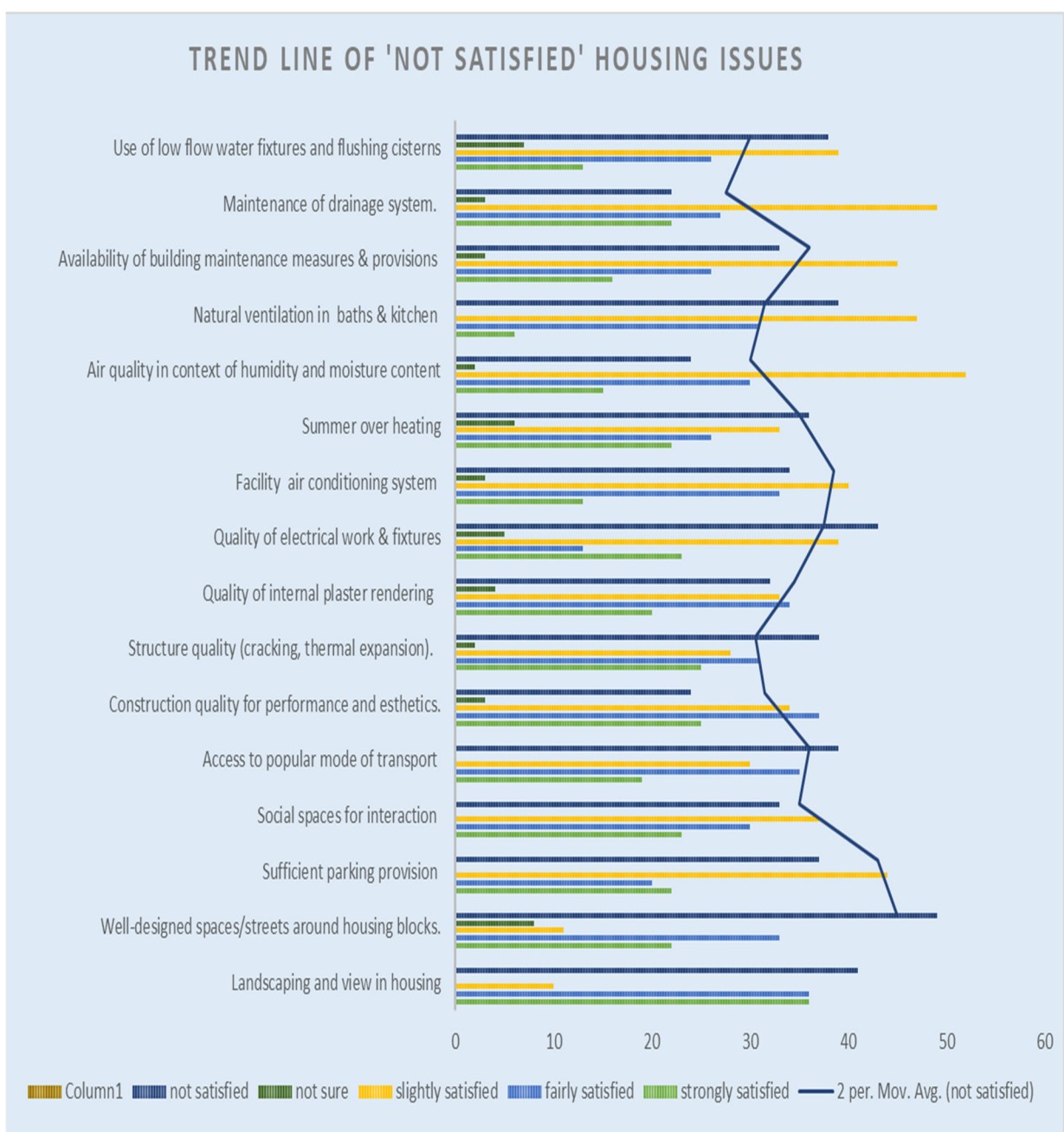

**Figure 6.** "Not satisfied" trend.

Figure 7 highlights the trend for "slight satisfaction" responses for different variables listed in the matrix. The analysis showed that six variables of the matrix attained a level of slight satisfaction, such as maintenance of drainage system, facility air conditioning system, air quality (humidity and moisture), construction quality, quality of internal plaster rendering, and parking provision.

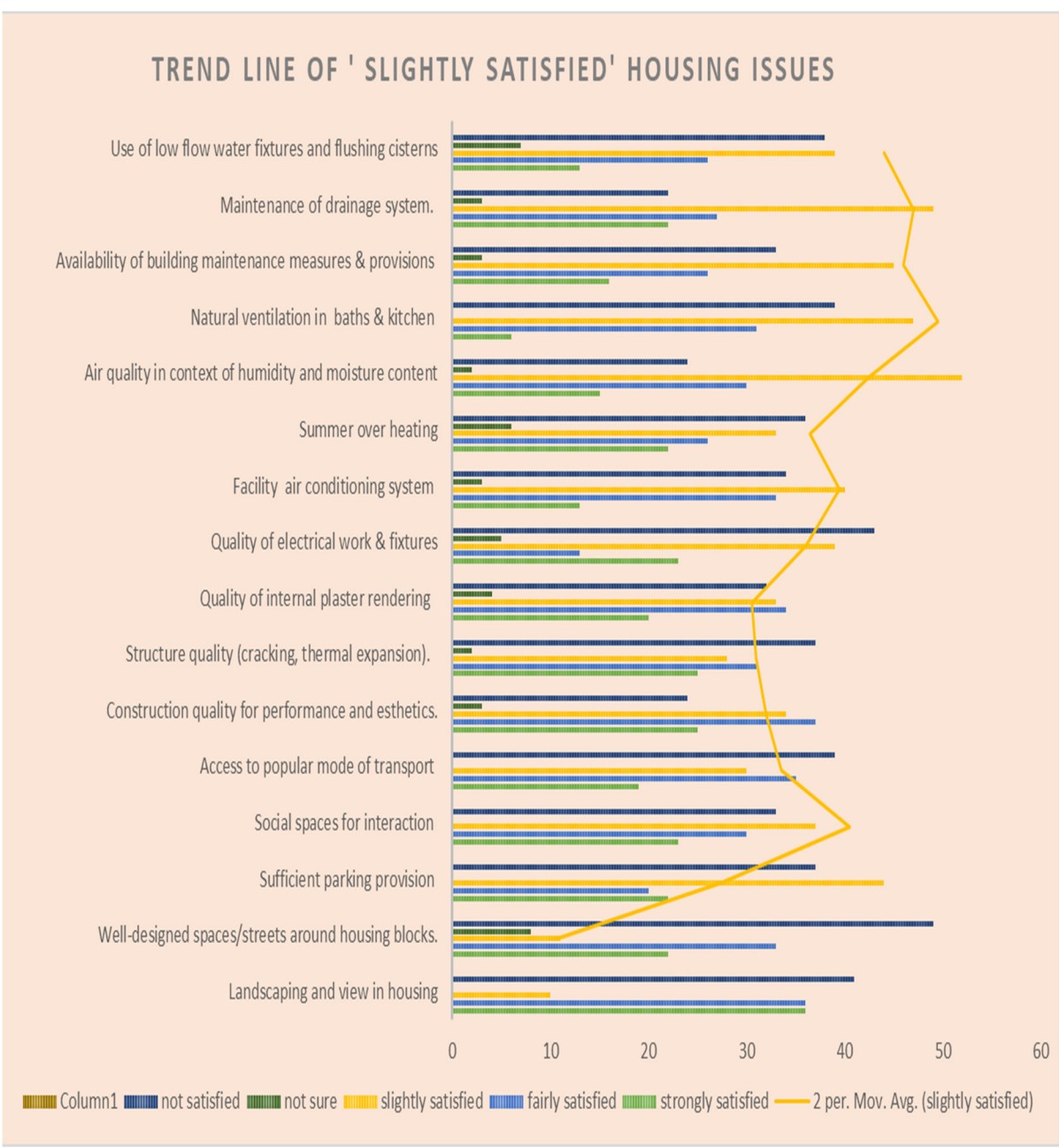

**Figure 7.** "Slightly satisfied" trend.

Figure 8 shows that at certain locations in Dubai, users showed strong contentment with the variables listed in the matrix. The same variables were met with discontentment in the other emirates of Sharjah and Ajman. Variables such as low-water flow fixtures, drainage system, maintenance, structural and construction quality, social spaces, well-designed streets, and landscaping views reached the threshold of satisfaction.

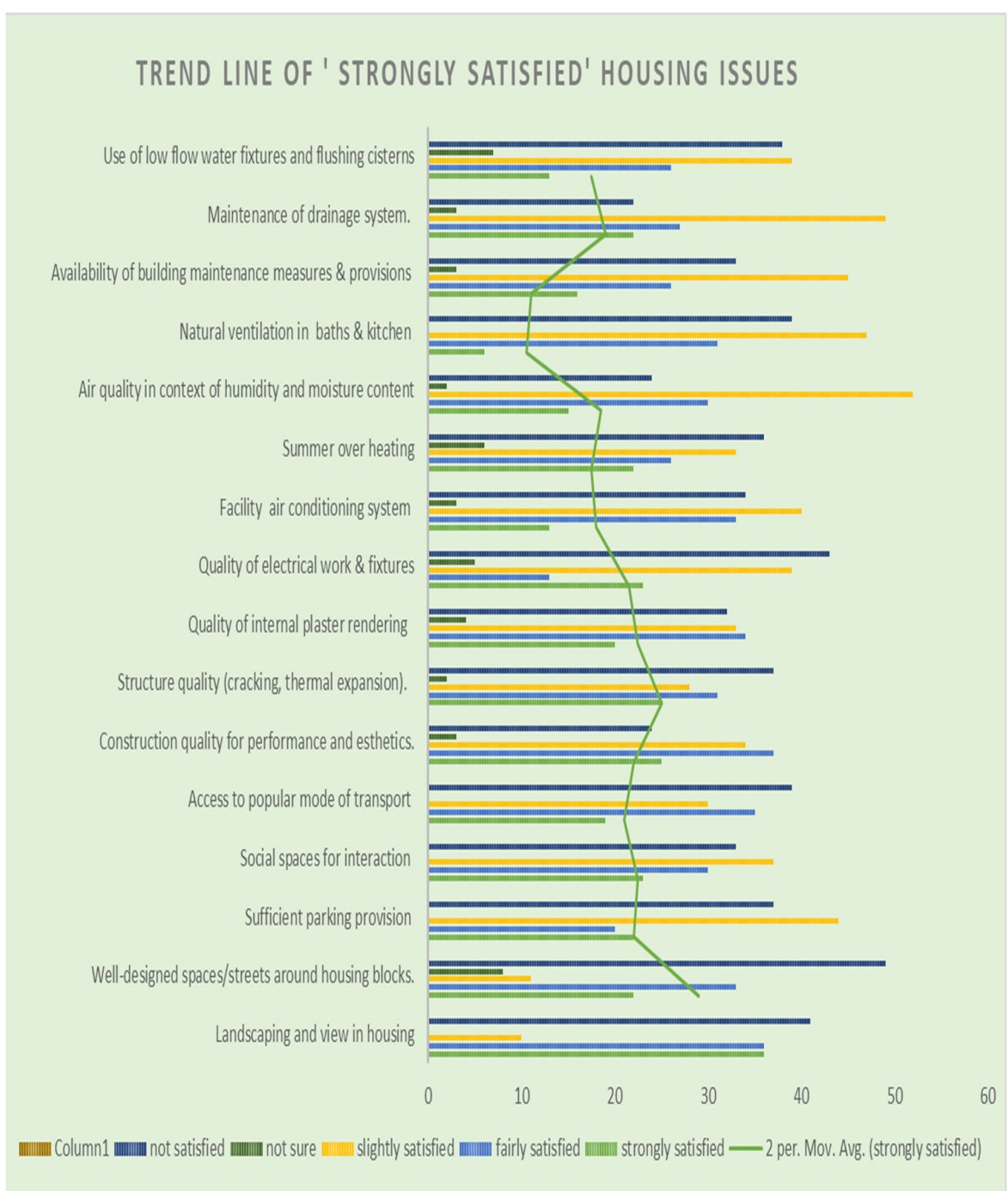

**Figure 8.** "Strongly satisfied" trend.

### 4. Conclusions

This study initiated the identification of poor design issues in affordable housing and set viable objectives to identify these issues in affordable housing in the UAE. Real estate and housing design in the UAE are highly diversified with impressive, quality buildings with amazing neighbourhoods and street networks, located close to edifices, particularly residential buildings with moderate to serious building issues.

Respondents from Dubai expressed high satisfaction with building design, function, and performance. However, data analysis from Sharjah and Ajman revealed a mixed trend of both satisfied and unsatisfied users. A large number of respondents were dissatisfied with building construction, services, defects, and maintenance.

The results of this study identified 16 poor design issues in UAE affordable housing. It is evident that most of the building defects identified are related to construction, structure, services, and maintenance. However, users of residential buildings in all three states showed satisfaction with architectural design. Consequently, these identified low/deficient design factors have established a link between building design issues and user satisfaction.

It is proposed that this study's end results (identified issues of poor design) can be used in three ways. First, they are a benchmark to improve the construction/services/design/structure of new affordable housing. Second, as a real estate tool, buyers and tenants can use these results to check the building conditions. Third, the end results can be referred to by building maintenance and management organisations to improve their periodic maintenance schedule to prevent breakdown or hazardous maintenance. This study concludes that deficient or poor design of affordable housing causes components to deteriorate, increases facility management, adds financial burden, and shortens building life cycles.

This study is an individual effort of an academician, and the scope of research was limited to aspects of residents' safety, welfare, and wellbeing. The end results of the study will add to the knowledgebase of freehold residential buildings and their issues but will not be a part of policy documents. However, lessons learned from data collection and analysis could be referred to through open access databases as "suggestions" concerning building control/regulation departments. In addition, the results of this study will provide a blueprint and strategy for triangular dialogue between regional building designers, construction managers, and end users to improve the design of new buildings. This study considers that, because of similarities in design and construction techniques, climate, culture, and social issues, the end results of this research could also be used for housing assessment in other GCC countries.

**Funding:** This research received no external funding.

**Institutional Review Board Statement:** Not applicable.

**Informed Consent Statement:** Not applicable.

**Data Availability Statement:** Not applicable.

**Acknowledgments:** The author is thankful to Ajman University UAE for providing research facilities and assistance in the publication of this study.

**Conflicts of Interest:** The author declares no conflict of interest.

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
