# Peer review of "Matrix of Affordable Housing Assessment: A Development Process"

_designs, 2021_

Round 1

Reviewer 1 Report

It is a rigorous and interesting research, which shows the ability of its authors to address a complex and multi-thematic observation.

I would like to make some main observations that could be considered for a more precise communication of the results.

1- First, the description of the research and its methodology are correct. However, a much deeper discussion is missed. The results are presented in a descriptive way, but it would be necessary to propose meaningful justified readings of their qualitative values. A direct connection is not established between the values ​​obtained and the possible reasons that would explain them.
2- Secondly, the results are offered in the form of repetitive tables, which are not very eloquent. It could be pertinent to use another model of visual graphics in order to have basic infographics of compared values, striking data or aspects that allow the detection of explicit causes or consequences.
3- Thirdly, a spectrum of housing issues that affect very heterogeneous areas is addressed. One might wonder to what extent some questions are difficult to answer without an explanation that clarifies their interpretation (such as number A1 or A2); other issues do not seem to provide critical value (such as B1, B2, B6). The critical sense of several of the questions is not very clear either, or some contradictions are detected (it is asked in a positive way about the existence of sufficient parking, but then questions about sustainability are included that cover very few aspects). Perhaps it would be convenient to indicate explicitly which are the bibliographic references in Table 1 that are used for each package of questions. Other references that might be helpful:
- Brkanić, Ivana. "Housing quality assessment criteria." Electronic Journal of the Faculty of Civil Engineering Osijek-e-GFOS 8.14 (2017): 37-47.
- Montaner, Josep Maria; Muxi, Zaida, and Falagan, David H . "Tools for Inhabiting the Present: Housing in the 21st Century". Polytechnic Foundation of Catalonia, 2011.
4- The relationship between affordability and the questionnaire model is somewhat blurred. It would be useful to strengthen the definition of affodability that is being considered and how this affects the questionnaire template. Some references from other contexts could be:
- Chan; Albert PC; and Michael Atafo Adabre. "Bridging the gap between sustainable housing and affordable housing: The required critical success criteria (CSC)." Building and environment 151 (2019): 112-125.
- Falagan, David H. "Innovation in Affordable Housing." Barcelona 2015-2018 (Ajuntament de Barcelona, ​​2019).
-Esruq-Labin, A. M. J., et al. "Criteria for affordable housing performance measurement: a review." E3S web of conferences. Vol. 3. EDP Sciences, 2014.

It would be convenient to review all these aspects to obtain an article whose scientific value can be used to detect areas for improvement in the production of affordable housing.

Author Response

Dear Reviewers

Please see attached files

1- Revised Paper

2- Author's response to comments

All comments are accepted and revised. 

Thanks 

Reviewer 2 Report

The text refers to a survey on the understanding of affordable housing in a specific region. This is a relevant study, however, some aspects should be highlighted to be improved if the author so understands:
1. Reference should be made to the typology of reference buildings (are they mostly collective housing, detached houses, etc?);
2. The qualitative and quantitative criteria of the sample are not explicit (123 responses to the survey);
3. The use of images of some of the buildings targeted by the survey (drawings or photographs) would be very useful for a more accuracy reading;
4. In conclusion, it would be very enriching to reflect and point out paths on some aspects that stand out in the survey, namely:
a) Implementation of passive design strategies in buildings;
b) Or, the use of sustainable materials and their relationship with local vernacular architecture.

Author Response

Dear Reviewer 

Please refer the attachment 

Thanks 

Dr. Afaq Hyder

Reviewer 3 Report

What do you mean important? is there an unimportant country in GCC? "United Arab Emirates (UAE) is an important country of Gulf Cooperation Countries (GCC), it is a unique because of its multinational harmonized culture."

The text needs careful language editing, see the corrections just in the first sentence!

The structure of the paper is not easy to follow, requires improvements. 

Tables 5-20 can be combined concisely.

Conclusions should be supported with possible implementations!

Author Response

(The authors gave the same response as above.)

Round 2

Reviewer 1 Report

Comments and suggestions from report 1 were considered and important improvements were incorporated to the paper.

This reviewer consider the paper is ready to be accepted.

Reviewer 3 Report

the structure of the paper is still not up to standards. I suggest checking the journal guidelines and reorganizing them.